# Occurrence and Identification of Yeasts in Production of White-Brined Cheese

**DOI:** 10.3390/microorganisms10061079

**Published:** 2022-05-24

**Authors:** Athina Geronikou, Nadja Larsen, Søren K. Lillevang, Lene Jespersen

**Affiliations:** 1Department of Food Science, University of Copenhagen, Rolighedsvej 26, 1958 Frederiksberg, Denmark; athina.geronikou@food.ku.dk (A.G.); lj@food.ku.dk (L.J.); 2Arla Innovation Centre, Agro Food Park 19, 8200 Aarhus, Denmark; sklv@arlafoods.com

**Keywords:** white-brined cheese, spoilage yeasts, yeast taxonomy, dairy production

## Abstract

The aim of this study was to reveal the sites of yeast contamination in dairy production and perform taxonomic characterization of potential yeast spoilers in cheese making. Occurrence of spoilage yeasts was followed throughout the manufacture of white-brined cheese at a Danish dairy, including the areas of milk pasteurization, curd processing, and packaging (26 sites in total). Spoilage yeasts were isolated from whey, old cheese curd, and air samples in viable counts of 1.48–6.27 log CFU/mL, 5.44 log CFU/g, and 1.02 log CFU/m^3^, respectively. Yeast isolates were genotypically classified using (GTG)_5_-PCR fingerprinting and identified by sequencing of the D1/D2 region of the 26S rRNA gene. The largest yeast heterogeneity was found in old curd collected under the turning machine of molds, where 11 different yeast species were identified. The most frequently isolated yeast species were *Candida intermedia*, *Kluyveromyces marxianus*, and *Pichia kudriavzevii*. The less abundant yeast species included *Candida auris*, *Candida parapsilosis*, *Candida pseudoglaebosa*, *Candida sojae*, *Cutaneotrichosporon curvatus*, *Cutaneotrichosporon moniliiforme*, *Papiliotrema flavescens*, *Rhodotorula mucilaginosa*, *Vanrija humicola*, and *Wickerhamiella sorbophila*. The awareness on occurrence and taxonomy of spoilage yeasts in cheese production will contribute to a knowledge-based control of contaminating yeasts and quality management of cheese at the dairies.

## 1. Introduction

White-brined cheeses, such as Feta, Domiati, and Halloumi, originate from the Mediterranean region and the Middle East [1]. Different varieties of white-brined cheese have traditionally been made from ovine, caprine, bovine or cow milk, and nowadays, they are industrially produced or at an artisanal scale in Europe, Turkey, Northern Africa, and some regions of Asia and South America. The main production steps of white-brined cheese involve milk pasteurization, addition of starter cultures and rennet, followed by milk coagulation and curd formation. Subsequently, the curd is drained, salted, and ripened into brine to obtain the final product [2,3,4].

Yeasts are the major spoilage microorganisms in white-brined cheeses. Depending on the species, yeasts can readily utilize milk carbohydrates, such as lactose and galactose, organic acids, and proteins, and grow at dairy-relevant conditions, i.e., refrigerated temperatures, acidic pH, reduced water activity, and high salt concentrations [5,6,7]. The most common yeast spoilers in white-brined cheeses belong to the genera *Candida*, *Cryptococcus*, *Debaryomyces*, *Geotrichum*, *Kluyveromyces*, *Pichia*, *Rhodotorula*, *Saccharomyces*, *Torulospora*, *Trichosporon*, *Yarrowia*, and *Zygosaccharomyces* [7,8,9]. When grown in high numbers (typically 5–6 log CFU/g), the enzymatic activities of yeasts lead to various quality defects, such as off-flavors, discoloration, the swelling of the cans, and the softening of the cheese texture [1,5,7,9,10].

The common sources of yeast contamination in dairy plants include production facilities, raw materials, brine, air, wooden shelves of ripening, and personnel [5,6,7,8,11]. Yeast occurrence is mostly attributed to their survival through sanitizers, as well as cross-contamination via processing environment and air [6,9,12]. Various studies have demonstrated that aerosols, in form of droplets or dust, can be easily dispersed by air flow, introducing a major way of yeast transmission in dairy environment [13]. Stobnicka-Kupiec et al. [14] identified *Candida* spp., *Cryptococcus* spp., *Debaryomyces hansenii*, *Geotrichum candidum*, *Rhodotorula* spp., and *Yarrowia lipolytica* in the air and surface samples of commercial and traditional Polish dairy plants (with expertise in milk, butter, cream, yoghurt, and cheese production; not including mold-ripened cheese), where the highest concentration of yeasts and molds was found on the worktops in the milk reception and cheese production areas and in the air samples. An earlier study revealed that the major fungal loads in the air of outdoor and indoor locations of a Greek dairy plant was presented by *Cladosporium* spp., *Penicillium* spp., and some unidentified yeasts [15].

Traditional dairy products are especially prone to microbial contamination, mostly due to the usage of unpasteurized milk [1,14,16]. At commercial dairies, production processes are based on specific management systems, including GMP (Good Manufacturing Practices), GHP (Good Hygiene Practices), SOP (Standard Operating Practices), and the universal Hazard Analysis and Critical Control Point (HACCP) system, targeting prevention and control of food contamination [9,14,15,17,18]. Despite of the implementation of the quality-management systems, yeast spoilage is still of concern in the dairy industry.

Identification of yeasts at species and strain level is essential to trace the ways of yeast contamination in dairy production and evaluate their spoilage potential. Most of the published studies revealed the spots of contamination at specific dairies and quantified the yeast loads, while only a few of them proceeded with taxonomic characterization of the isolated yeasts [14,16,17]. The aim of this study was therefore to reveal the hotspot areas of yeast contamination throughout production of white-brined cheese at a Danish dairy and characterize the taxonomic diversity of potential yeast spoilers.

## 2. Materials and Methods

### 2.1. Sample Collection

Sampling was performed during two visits at a Danish dairy production of white-brined cheese, in November 2019 (Trial 1) and January 2020 (Trial 2), referred to as T1 and T2, respectively (Table 1 and Table 2). In T1, 47 samples were collected from 24 different sites in production of white-brined cheese including milk, curd, cheese, brine, ingredients, air, and swab from the cheese cutter. The second visit (T2) was mostly carried out as a supplementary sampling at the hotspots areas revealed in T1. In total, 22 samples from 11 different sites were collected in T2 including curd, whey, cheese, and air.

Figure 1 presents the scheme of cheese production line with highlighted sampling sites and sample codes. The production area (A) refers to the cheese vats, (B) and (C) to the mechanical tunnel and draining room, respectively, and (D) and (E) to the packaging areas. Major production steps included milk pasteurization (Sample A1), followed by the addition of rennet and starter cultures, which allowed milk coagulation and curd formation (Sample A2). Afterwards, the curd was added into molds (Samples B3 and C1), drained for 20–26 h at 13 °C (Sample C2), and cut into cubes (Sample D5). The cubes were immerged into 12% (*w*/*v*) brine for 12–72 h at 5 °C (Samples D7 and D8). Finally, the cubes were mixed with other ingredients such as oil, vegetables, and herbs (Samples E1, E2, E3, and E5), filled in “consumer pack” containers (Sample E4), and finally transported to the storage room until their distribution.

### 2.2. Enumeration and Isolation of Yeasts

Solid samples of curd, cheese, and ingredients in amounts of 10 g were mixed with 90 mL of Saline Peptone Diluent (SPO) (0.1% *w*/*v* peptone, 0.03% *w*/*v* Na_2_HSO_4_, 1% *w*/*v* NaCl, pH 5.6 ± 0.1) and homogenized in a Stomacher Bags Mixer (InterScience, Saint Nom la Bretêche, France) for 180 s at high speed. Whey samples were additionally incubated at 25 °C for 48 h, before yeast enumeration, as an enrichment step to promote yeast growth. The swab sample from the cube cutter (15 × 15 mm cube size, random sampling in different parts of the cube cutter) was collected using Compact Dry Swab (HyServe, Kirchheim bei München, Germany) in 1 mL SPO added 1% (*w*/*v*) NaCl. Air samples of 500 L were collected with an air sampler Sampl’air Lite (AES Laboratoire, BioMérieux, Marcy-l’Étoile, France). Serial dilutions of the samples were prepared with Pro-Media MT-11PBS diluent (Elmex, Tokyo, Japan), and plated in duplicates on Malt Yeast Glucose Peptone (MYGP) agar (0.3% *w*/*v* malt extract, 0.3% *w*/*v* yeast extract, 1% *w*/*v* D-Glucose monohydrate, 0.5% *w*/*v* peptone, 2% *w*/*v* agar, 1% *w*/*v* NaCl, supplemented with 100 mg/L of chloramphenicol and 50 mg/L of chlortetracycline, pH 5.6 ± 0.1). For CFU enumeration, plates with 20–300 colonies were selected and the results are presented as average values of log_10_ and their standard deviations are calculated. For brine analysis, SPO and MYGP agar, supplemented with 4% (*w*/*v*) NaCl, were used in order to mimic the brine environment. Plates were incubated under aerobic conditions at 25 °C for 3–5 days. Representative colonies, randomly selected (10–20 colonies depending on the CFU counts), were purified by streaking on MYGP agar plates and grouped based on their micro- and macromorphological characteristics (surface, margin, profile, size, and color). The number of isolates for the subsequent rep-PCR (Repetitive Extragenic Palindromic Polymerase Chain Reaction) analysis was calculated as the square root of the number of colonies from each colony group. In case of low counts (Samples B6 and D1), all colonies were isolated and purified. Selected isolates were grown in MYGP broth overnight and stored at −80 °C in 20% *v*/*v* glycerol. All reagents were purchased from Sigma A/S (Søborg, Denmark) or Merck (Søborg, Denmark), unless otherwise specified.

### 2.3. Molecular Characterization of Yeast Isolates

#### 2.3.1. Rep–PCR

The total yeast DNA was extracted from the colonies grown on MYGP agar plates using the InstaGene Matrix DNA extraction kit (Bio-Rad Laboratories, Hercules, CA, USA). Rep-PCR was conducted in a 25 μL volume mixture containing 13 μL of Taq DNA Polymerase 2× Master Mix RED (Ampliqon, Odense, Denmark), 5 μL Primer GTG_5_ (Integrated DNA Technologies, Denmark), 4 μL sterile Milli-Q water, and 3 μL DNA. The PCR reaction was carried out in a SureCycler 8800 thermocycler (Agilent Technologies, Santa Clara, CA, USA) using the following program: initial denaturation for 7 min at 95 °C, followed by 30 cycles of 95 °C for 1 min, 45 °C for 1 min, and 65 °C for 8 min, and an elongation step of 65 °C for 16 min. The rep-PCR products were separated by 1.5% agarose gel electrophoresis (5 h, 120 V) in Tris-Borate-EDTA buffer (0.5 × TBE), using an O’GeneRuler 1 kb DNA ladder (Thermo Scientific, Roskilde, Denmark) as a reference marker. The rep-PCR profiles were clustered using Bionumerics 7.1 software (Applied Maths, BioMérieux, Schaerbeek, Belgium) based on Dice’s Coefficient of similarity with the Unweighted Pair Group Method and Arithmetic mean clustering algorithm (UPGMA).

#### 2.3.2. 26S rRNA Gene Sequencing

Sequencing of the D1/D2 region of the 26S rRNA gene was performed for all the isolates clustered (39 in total) using the primers NL-1 (5′-GCA TAT CAA TAA GCG GAG GAA AAG-3′) and NL-4 (5′-GGT CCG TGT TTC AAG ACG G-3′), as described by Van Der Aa Kühle and Jespersen [19]. Shortly, the PCR reaction was conducted in a 50 μL volume mixture containing 25 μL of Taq DNA Polymerase 2× Master Mix RED (Ampliqon, Odense, Denmark), 5 μL Primer mix NL-1 and NL-4, 17 μL sterile Milli-Q water, and 3 μL of the total DNA from yeast. The PCR reaction was carried out in a SureCycler 8800 thermocycler (Agilent Technologies, Santa Clara, CA, USA), using the following program: initial denaturation for 5 min at 95 °C, followed by 30 cycles at 95 °C for 90 s, 53 °C for 30 s, and 72 °C for 90 s, and the final elongation step at 72 °C for 7 min. The DNA sequencing (using the same primers; NL1 and NL4) was performed by Macrogen (Amsterdam, The Netherlands). Sequences were manually corrected, assembled with CLC Genomics Workbench version 7.9.1 software (QIAGEN Digital Insights, Redwood City, CA, USA), and compared to the reported 26S rRNA gene sequences in GenBank using the Basic Local Alignment Search Tool (BLAST) algorithm. The nucleotide sequences have been deposited in GenBank under Accession Numbers OL744629–OL744667, as indicated in Table 3.

#### 2.3.3. Sequencing of the 5.8S rDNA-ITS Region

Sequencing of the Internal Transcribed Spacer (ITS) (ITS1-5.8S rDNA-ITS2 region) was additionally performed to differentiate closely related species of *Kluyveromyces marxianus* and *Kluyveromyces lactis* that were not differentiated by sequencing the D1/D2 region of the 26S rRNA gene [20]. For amplification of the 5.8S-ITS fragment, primers ITS-1 (5′-TCC GTA GGT GAA CCT GCG G-3′) and ITS-4 (5′-TCC TCC GCT TAT TGA TAT GC-3′) were used as previously described [19,21,22]. Shortly, the PCR reaction was conducted in a 50 μL volume mixture containing 25 μL of Taq DNA Polymerase 2× Master Mix RED (Ampliqon, Denmark), 5 μL Primer mix ITS-1 and ITS-4, 17 μL sterile Milli-Q water and 3 μL of samples’ DNA. The PCR reaction was carried out with a SureCycler 8800 (Agilent Technologies, Santa Clara, CA, USA) under the following conditions: 3 min of initial denaturation at 95 °C, followed by 30 cycles of 95 °C for 30 s, 60 °C for 40 s, 72 °C for 30 s, and the final elongation step at 72 °C for 10 min. The PCR products were treated as in Section 2.3.2. of this publication.

### 2.4. Phenotypic Tests 

The isolates of *Kluyveromyces* spp. (6 isolates in total) were tested for assimilation and fermentation of glucose, lactose, maltose, galactose, sucrose, and raffinose to distinguish the species of *K. marxianus* and *K. lactis* [23]. Isolates were grown on MYGP agar for 3 days at 25 °C before the experiments. For fermentation tests, 5 mL medium (0.45% *w*/*v* yeast extract; 0.75% *w*/*v* peptone; 12 mL Bromothymol blue solution; and up to 1000 mL distilled water, pH 6.0 ± 0.2) was added into Durham tubes. The medium for assimilation tests (0.067% *w*/*v* Difco Yeast Nitrogen Base, 0.05% *w*/*v* Carbohydrate, pH 5.6 ± 0.1) was added in a volume of 0.5 mL into the respective test tubes. Solutions of glucose, lactose, maltose, galactose, sucrose (6% *w*/*v* each), and raffinose (12% *w*/*v*) were distributed in the tubes for fermentation (2.5 mL) and assimilation (0.5 mL) tests, and afterwards, yeast-culture suspension of 0.1 mL (yeast colony material mixed with 4.5 mL sterile water) was transferred into the test tubes. The tubes were incubated at 25 °C for 4 weeks and checked weekly for gas production and/or change of the color (fermentation test) and using a Wickerhams card (assimilation test) [23]. Strains of *Kluyveromyces marxianus* CBS 1553 and *Kluyveromyces lactis* CBS 845 were obtained from the CBS culture collection (Centraalbureau voor Schimmelcultures, CBS, The Netherlands) and used as positive controls.

## 3. Results and Discussion

### 3.1. Yeast Viable Counts

The viable counts of yeasts in milk (2 samples), whey (4 samples), curd (18 samples), cheese (24 samples), and brine (2 samples) are presented in Table 1. The levels of yeasts in raw materials (6 samples), air (12 samples), and swab (1 sample) are shown in Table 2. The highest yeast counts of 5.44 ± 1.07 log CFU/g were detected in the old curd under the turning machine (sample B4). Additionally, yeasts were found in recirculated whey (separated from cheese fines) at levels of 1.48 ± 0.01 log CFU/mL (sample B7b, before the enrichment step) and 6.27 ± 0.01 log CFU/mL (sample B8b, after the enrichment step). Yeast concentration in the air from the draining room (sample C3) was 1.02 ± 0.12 log CFU/m^3^. The yeast counts in all other production samples were below the detection limit (less than 2 log CFU/g for solid samples or less than 1 log CFU/mL, m3 or m2 for liquid, air, and swab samples, respectively). Despite undetectable levels of yeast counts, especially in the final product (E4), possibility of yeast contamination and subsequent cheese spoilage during storage cannot be excluded. Air, cheese curd, and whey have previously been recognized as the major sources of yeast contamination in dairy production. Yeast counts in curd under the turning machine in this study were close to the values reported for curd in Pecorino Crotonese cheese manufacture (3.7 log CFU/g) and Bryndza cheese produced from raw ewe milk (4.1–6.2 log CFU/g) [12,24]. Besides, the CFU counts in recirculated whey were lower than the numbers reported for whey from production of artisanal white-pickled cheese (2.5–5.6 log CFU/mL) [4,25]. The absence of yeast contamination in brine in this study is in contrast with the results reported for white-pickled and Feta cheeses (5–7 log CFU/mL) [4,25]. Discrepancies between the results can be explained by the use of fresh brine, in this study (6 weeks old with an expiration period of at least 6 months), as well as by differences in production processes and hygiene conditions between the dairies.

Supporting our results, microbial air loads of 1–3 log CFU/m^3^ have commonly been reported in dairy plants [18,26,27]. Until now, there is no legislation concerning the limits of airborne contaminants in dairy production, though several scientific recommendations have been published [28,29,30,31]. Thus, according to the American Public Health Association (APHA), the fungal loads in dairy facilities should not exceed the range of 1.8–2.6 log CFU/m^3^ [30,31]. Consequently, the levels of yeasts in the air samples in this study are in accordance with the recommended limits. However, there is no doubt that the contamination from air should be avoided, e.g., by prevention of aerosols caused by high pressure flushing.

### 3.2. Phylogenetic Characterization of Yeasts

In total, 99 morphologically distinct isolates were purified from 69 samples in the white-brined cheese production. Among them, 39 isolates were clustered using rep-PCR and further identified by sequencing the D1/D2 region of the 26S rRNA gene and the 5.8S rDNA-ITS region. Figure 2 presents the phylogenetic dendrogram of these isolates based on the rep-PCR fingerprints. Using a similarity cut-off of 85%, the isolates were grouped into 13 clusters. Major clusters were comprised of species *Pichia kudriavzevii* (formerly *Issatchenkia orientalis*, anamorph *Candida krusei*) (Cluster 3), *Candida intermedia* (Cluster 8), and *K. marxianus* (anamorph *Candida kefyr*) (Cluster 10). The *Candida* spp., *K. marxianus*, *P. kudriavzevii*, and *Wickerhamiella sorbophila* (syn. *Candida sorbophila*) have been assigned to the division of *Ascomycota*, class of *Saccharomycetes*. *Candida* was the most diverse genus in this study, represented by five species, namely *Candida auris*, *C. intermedia*, *Candida parapsilosis*, *Candida pseudoglaebosa*, and *Candida sojae*. 

Other identified species, *Cutaneotrichosporon curvatus* (formerly *Candida curvata*, syn. *Cryptococcus curvatus*), *Cutaneotrichosporon moniliiforme* (formerly *Trichosporon moniliiforme*), *Paliliotrema flavescens* (formerly *Cryptococus flavescens*), *Rhodotorula mucilaginosa*, and *Vanrija humicola* (formerly *Cryptococcus humicola*) belong to the division of *Basidiomycota*, classes of *Microbotryomycetes* and *Tremellomycetes*. The taxonomic identities of the isolates from production samples and their GenBank Accession Numbers are listed in Table 3. The obtained sequences were annotated to 13 yeast species showing 99.7–100% similarity to the reference GenBank sequences.

### 3.3. Phylogenetic Characterization of Yeasts

In total, 99 morphologically distinct isolates were purified from the white-brined cheese production. Among them, 39 isolates were clustered using rep-PCR and further identified by sequencing the D1/D2 region of the 26S rRNA gene and the 5.8S rDNA-ITS region. Figure 2 presents the phylogenetic dendrogram of these isolates based on the rep-PCR fingerprints. Using a similarity cutoff of 85%, the isolates were grouped into 13 clusters. Major clusters were comprised of species *Pichia kudriavzevii* (formerly *Issatchenkia orientalis*, anamorph *Candida krusei*) (Cluster 3), *Candida intermedia* (Cluster 8), and *K. marxianus* (anamorph *Candida kefyr*) (Cluster 10). *Candida* spp., *K. marxianus*, *P. kudriavzevii*, and *Wickerhamiella sorbophila* (syn. *Candida sorbophila*) were assigned to the division of *Ascomycota*, class of *Saccharomycetes*. *Candida* was the most diverse genus in this study, represented by five species, namely *Candida auris*, *C. intermedia*, *Candida parapsilosis*, *Candida pseudoglaebosa*, and *Candida sojae*. Other identified species, *Cutaneotrichosporon curvatus* (formerly *Candida curvata*, syn. *Cryptococcus curvatus*), *Cutaneotrichosporon moniliiforme* (formerly *Trichosporon moniliiforme*), *Paliliotrema flavescens* (formerly *Cryptococus flavescens*), *Rhodotorula mucilaginosa*, and *Vanrija humicola* (formerly *Cryptococcus humicola*) belong to the division of *Basidiomycota*, classes of *Microbotryomycetes* and *Tremellomycetes*. The taxonomic identities of the isolates from production samples and their GenBank Accession Numbers are listed in Table 3. The obtained sequences were annotated to 13 yeast species, showing 99.7–100% similarity to the reference GenBank sequences.

Due to close genotypic relatedness, the species *K. marxianus* and *K. lactis* could not be distinguished by 26S rRNA gene sequencing in this study. Based on the sequence analysis of the 5.8S rDNA-ITS region, the isolates were annotated to *K. marxianus* with 100% homology to GenBank. Additionally, phenotypic tests showed that the isolates could readily utilize and ferment glucose, galactose, sucrose, lactose, and weakly raffinose (Appendix A). Concurrently, no fermentation of maltose was observed during the 4-week incubation period, indicating that all *Kluyveromyces* isolates belong to *K. marxianus* [32].

### 3.4. Diversity of Yeasts in Cheese Production

The predominant species in white-brined cheese production were *P. kudriavzevii* (26% of total isolates), *C. intermedia* (20% of total isolates), and *K. marxianus* (15% of total isolates) identified from whey samples (B7b, B8b), old curd (B4), cheese on conveyor belt (D1), and air (B6). The largest abundance and diversity of yeast species (64% of the isolates) was found in the old curd (B4), presenting a nutrient rich substrate for yeast growth (Table 3). Species *C. intermedia* was predominantly found in the old curd (24% of curd isolates) and prevailed in cheese samples from conveyor belt (D1). Other species, isolated from the old curd, were *C. auris* (5%), *C. pseudoglaebosa* (5%), *C. sojae* (5%), *C. curvatus* (3%), *P. flavescens* (5%), *R. mucilaginosa* (3%), *V. humicola* (5%), and *W. sorbophila* (3%) (Table 3). Species *C. parapsilosis* (50%) and *C. moniliiforme* (50%) were identified in the air from the draining room (C3). Among them, the species of *C. intermedia*, *K. lactis*, *P. kudriavzevii*, and *R. mucilaginosa* are well-known contaminants in white-brined cheese [7,33,34,35].

It has been reported that *Candida* spp., such as *C. parapsilosis* and *C. sojae*, can be easily distributed through the dairy-processing environment and air [6,14]. In accordance with this study, predominance of *C. intermedia* has been demonstrated in the cheese-production environment [36], as well as in various types of cheeses, such as Serro Minas [7], Pecorino Crotonese [12], and cheese brines [37,38]. *C. sojae* has occasionally been detected in yoghurts [6], while *C. pseudoglaebosa* has been found in raw milk from dairy farms in France [39,40]. In contrast, species *C. auris* is not of dairy origin but rather referred to as a human pathogen frequently isolated from the human body fluids and tissues [41,42]. Thus, curd contamination with *C. auris* in this study would probably rise from direct contact with personnel at the dairy.

*C. parapsilosis* was a rare species in this study, detected only in the air from the draining room (1 out of 2 isolates). It is a common spoilage organism in various types of cheeses (brined, ripened, Swiss-type, blue-veined), characterized by high proteolytic and lipolytic activities [7,11,43]. Similar to *C. auris*, *C. parapsilosis* is considered as an opportunistic human pathogen causing invasive candidiasis [11,44,45]. At the same time, *C. parapsilosis* is regularly encountered in other fermented products, having positive impact on their organoleptic properties. To the best of our knowledge, species of *C. auris* and *C. parapsilosis* have not been linked to any outbreak of infection.

Along with the old curd, *K. marxianus* and *P. kudriavzevii* were the major species detected in whey samples before and after the enrichment step (B7b and B8b). Furthermore, *K. marxianus* was identified in the air sample (B6). Species *P. kudriavzevii* is a common inhabitant of various fermented products, able to grow at low pH and high salt concentration [46,47]. Predominance of *K. marxianus* can be due to its ability to ferment residual lactose in cheese [20,48], and assimilate lactate produced by lactic-acid bacteria [49,50,51]. High prevalence of these species in white-brined cheese production is in agreement with several studies. Both *K. marxianus* and *P. kudriavzevii* have been reported as predominant yeast contaminants at different stages of May Bryndza cheese production from dairies in Slovakia and in Portuguese Serpa cheese [24,52,53]. In addition, *K. marxianus* was identified in Serro Minas and water buffalo Mozzarella cheeses [7,54], while *P. kudriavzevii* has previously been isolated from British Wensleydale cheese [11] and fresh curd of artisanal Canastra cheese [43].

Other less frequent species from white-brined cheese production were *C. curvatus*, *P. flavescens*, and *V. humicola* detected in old curd (B4), and *C. moniliiforme* in the air from the draining room (C3). These species formerly belonged to the genus *Cryptococcus*, but after a recent reclassification they were assigned to the mentioned genera [55]. In accordance with this study, a few incidences of *C. curvatus* have been reported in the air of dairy environments [56], whereas *P. flavescens* has rarely been identified in dairy matrices such as ice cream and cheese brine [38,45]. The presence of *C. moniliiforme* is undesirable in white-brined cheese, as it is able to form melanin-like brown pigments, leading to product discoloration [57]. *Cutaneotrichosporon* spp., *P. flavescens*, and *V. humicola* are not typical dairy spoilers, but rather found in various natural environments (i.e., plants, bird feces, and tree hollows) [58]. Therefore, their ability to grow in white-brined cheese needs further investigation.

The species *R. mucilaginosa* and *W. sorbophila* were detected in low frequency in cheese curd. Similarly, *W. sorbophila* has been found in low numbers in buffalo Mozzarella cheese [54], while *R. mucilaginosa* has occasionally been isolated from production of various cheeses (e.g., Serro Minas, Bryndza cheese) [7,52]. Furthermore, *R. mucilaginosa* is considered an emerging opportunistic pathogen causing fungemia in humans [59,60]. Likewise with other *Rhodotorula* species, *R. mucilaginosa* may impair the cheese quality, giving rise to yellow/red pigmentation on the cheese surface attributed to carotenoid production (e.g., β-carotene, torulene, torularhobin) [61]. Thus, despite the low numbers, the effect of *R. mucilaginosa* on cheese shelf life and safety should not be underestimated.

## 4. Conclusions

A high diversity of yeast species (13 in total) was found in the production of white-brined cheese at a Danish dairy. Most of the yeast species (i.e., *C. intermedia*, *K. marxianus*, *P. kudriavzevii*) have been characterized as typical dairy spoilers, while the others (i.e., *C. auris*, *C. parapsilosis*, *Cutaneotrichosporon* spp., *P. flavescens*, *V. humicola*) originate from different sources, such as plants, humans, and other environments. The contamination of white-brined cheese and consequent proliferation of yeasts might lead to quality defects such as off-flavors, discoloration, blowing, etc. Nevertheless, it can be foreseen that not all yeast species will be able to proliferate during the maturation and storage of white-brined cheeses. The mapping of yeast hotspot areas in dairy production, as well as species identification, can be beneficially used by dairies to eliminate cross contamination and prevent spoilage of the products. Additional studies are required to evaluate the ability of yeasts to survive and propagate in the final product and their effect on cheese quality and shelf-life. 

## Figures and Tables

**Figure 1 microorganisms-10-01079-f001:**
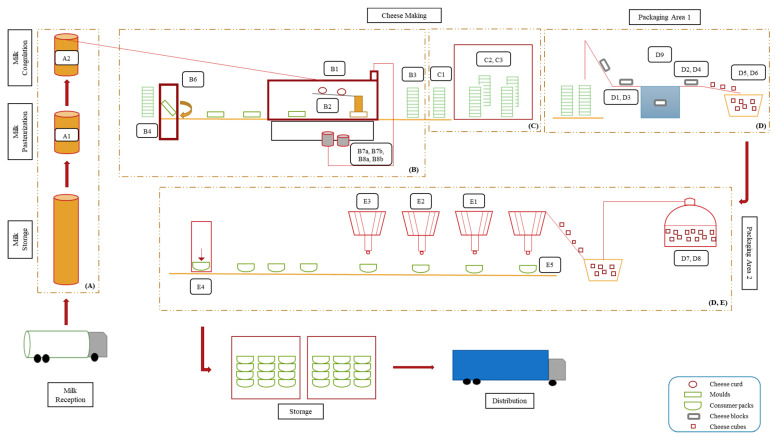
Schematic illustration of the white-brined cheese-production line, including (**A**) cheese vats, (**B**) mechanical tunnel, (**C**) draining room, (**D**,**E**) packaging areas. The sampling sites are denoted in boxes and described in Table 1 and Table 2.

**Figure 2 microorganisms-10-01079-f002:**
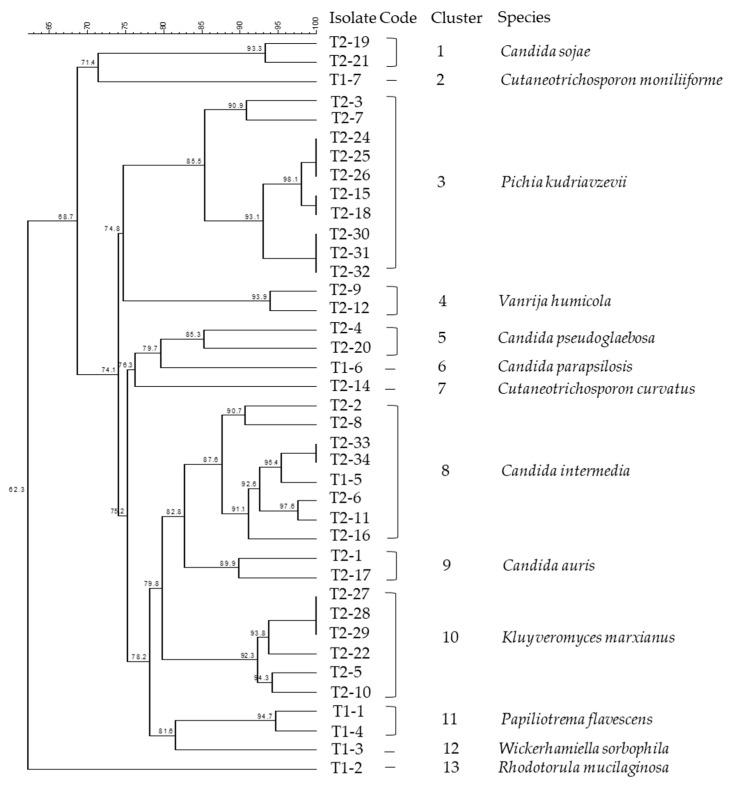
Dendrogram of (GTG)_5_-PCR fingerprints of yeast isolates collected from the production line of white-brined cheese. Clustering was based on Dice’s coefficient of similarity with the unweighted pair-group method with arithmetic average clustering algorithm (UPGMA). Numbers of nodes in the dendrogram denote the percentage of similarity between the clusters.

**Table 1 microorganisms-10-01079-t001:** Viable counts of yeasts in milk, whey, and cheese samples from white-brined cheese production.

Production Line	Sample ID ^a^	Trial	Sample Description	Log (CFU/g or CFU/mL) ^b^
Cheese vats (A)	A1	T1	Pasteurized milk before adding cultures	<2
	A2	T1	Cheese curd before pumping from the cheese vat into the mold	<2
Mechanical tunnel (B)	B1	T1	Cheese curd before drained off onto conveyor belt	<2
	B2	T1	Curd fines from saver	<2
	B3	T1	Cheese curd in molds	<2
	B4	T1, T2	Old cheese curd under the turning machine of molds	5.44 ± 1.07
	B7a	T2	Whey after pumping (Fresh)	<1
	B7b	T2	Whey after pumping (Recirculated)	1.48 ± 0.01
	B8a	T2	Whey after pumping (Fresh) ^c^	<1
	B8b	T2	Whey after pumping (Recirculated) ^c^	6.27 ± 0.01
Draining room (C)	C1	T1	Cheese curd entering the draining room	<2
	C2	T1, T2	Cheese curd at package pH (stacks of molds leaving the draining room)	<2
Packaging (D, E)	D1	T1, T2	Cheese on conveyor belt before heat bath	<2
	D2	T1, T2	Cheese on conveyor belt after heat bath	<2
	D3	T1	Cheese surface before heat bath	<2
	D4	T1	Cheese surface after heat bath	<2
	D5	T1	Cheese cubes	<2
	D7	T1	Brine in tank	<1
	D8	T1, T2	Cheese cubes in brine	<2
	E1	T1	Ingredients: Vegetables	<2
	E2	T1	Ingredients: Spices	<2
	E3	T1	Ingredients: Oil	<1
	E4	T1, T2	Packaged cheese after welding (final product at t = 0)	<2
	E5	T1	Cheese cubes after the filling machine	<2

^a^ Sampling sites as shown in Figure 1. ^b^ Mean values ± SD of the total yeast counts from two Trials (T1 and T2). ^c^ Whey samples analyzed after incubation at 25 °C for 48 h (enrichment step).

**Table 2 microorganisms-10-01079-t002:** Viable counts of yeasts in environmental samples from white-brined cheese production.

Production Line	Sample ID ^a^	Trial	Sample Description	Log (CFU/m^3^ or CFU/m^2^) ^b^
Mechanical tunnel (B)	B6	T1, T2	Air sample close to the turning machine	<1
Draining room (C)	C3	T1, T2	Air sample	1.02 ± 0.12
Packaging (D)	D6	T1	Swab Test of cube cutter	<1
	D9	T1, T2	Air sample close to heat bath	<1

^a^ Sampling sites as shown in Figure 1. ^b^ Mean values ± SD of the total yeast counts from two Trials (T1 and T2).

**Table 3 microorganisms-10-01079-t003:** Identification of yeasts from the white-brined cheese production by sequencing the D1/D2 region of the 26S rRNA gene.

Sample Description	Species	Length D1/D2 Region, bp	Homology to GenBank, %	Identities Gen Bank	Isolate Code ^a^	GenBank Accession No
B4: Old curd under the turning machine of molds	*Candida auris*	557	99.4	476/479	T2-1	OL744636
		566	99.4	476/479	T2-17	OL744651
	*Candida intermedia*	533	100	533/533	T1-5	OL744633
		533	100	533/533	T2-2	OL744637
		533	100	533/533	T2-6	OL744641
		512	100	512/512	T2-8	OL744643
		514	100	514/514	T2-11	OL744646
		526	100	526/526	T2-16	OL744650
	*Candida pseudoglaebosa*	583	99.8	574/575	T2-4	OL744639
		549	99.8	548/549	T2-20	OL744654
	*Candida sojae*	561	100	561/561	T2-19	OL744653
		592	99.7	572/574	T2-21	OL744655
	*Cutaneotrichosporon curvatus*	564	100	564/564	T2-14	OL744648
	*Kluyveromyces marxianus* ^b^	547	100	547/547	T2-5	OL744640
		542	99.8	541/542	T2-10	OL744645
	*Papiliotrema flavescens*	555	100	555/555	T1-1	OL744629
		613	99.8	612/613	T1-4	OL744632
	*Pichia kudriavzevii*	564	100	564/564	T2-3	OL744638
		505	100	505/505	T2-7	OL744642
		571	100	571/571	T2-15	OL744649
		541	100	541/541	T2-18	OL744652
	*Rhodotorula mucilaginosa*	568	100	568/568	T1-2	OL744630
	*Vanrija humicola*	527	100	527/527	T2-9	OL744644
		593	99.8	592/593	T2-12	OL744647
	*Wickerhamiella sorbophila*	575	99.2	393/396	T1-3	OL744631
B6: Air sample close to turning machine	*Kluyveromyces marxianus* ^b^	551	100	551/551	T2-22	OL744656
B7b: Whey after pumping (Recirculated, t = 0 h, 25 °C)	*Pichia kudriavzevii*	579	100	579/579	T2-24	OL744657
		553	100	553/553	T2-25	OL744658
		562	100	562/562	T2-26	OL744659
B8b: Whey after pumping (Recirculated, t = 48 h, 25 °C)	*Kluyveromyces marxianus* ^b^	528	100	528/528	T2-27	OL744660
		552	99.8	551/552	T2-28	OL744661
		523	100	523/523	T2-29	OL744662
	*Pichia kudriavzevii*	574	100	574/574	T2-30	OL744663
		568	99.8	567/568	T2-31	OL744664
		564	100	564/564	T2-32	OL744665
D1: Cheese on conveyor belt before heat bath	*Candida intermedia*	521	100	521/521	T2-33	OL744666
		520	100	520/520	T2-34	OL744667
C3: Air sample in the draining room	*Candida parapsilosis*	585	99.7	583/585	T1-6	OL744634
	*Cutaneotrichosporon moniliiforme*	605	100	605/605	T1-7	OL744635

^a^ Isolates obtained in Trial 1 and Trial 2 are denoted by T1 or T2, respectively. ^b^ Isolates’ annotation to *K. marxianus* was confirmed by sequencing the 5.8S rDNA-ITS region.

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
