# Peer review of "Occurrence and Identification of Yeasts in Production of White-Brined Cheese"

_microorganisms, 2022, doi:10.3390/microorganisms10061079_

Round 1
Reviewer 1 Report
The manuscript entitled "Occurrence and identification of yeasts in production of white-brined cheese” investigates on a current topic of research regarding the yeast contamination in dairy production.
The research investigation is well set up and manuscript is well written and easy to follow and the results are of interest to contribute to a knowledge-based control of contaminating yeasts and quality management of cheese at the dairies.
There are only minor concerns which should be fixed.
The authors should be indicate the total number of yeast isolates from the total 69 samples.
These are only the purified It is correct? “In total, 99 morphologically distinct isolates were purified from the white-brined 227 cheese production.”
Author Response
Dear Editor and Reviewers,
We would like to thank you for your constructive comments on our submitted manuscript ‘Occurrence and identification of yeasts in production of white-brined cheese’ in MDPI Microorganisms Journal.
Regarding the corrections (attached the manuscript with the tracking changes):
Com#1: The authors should be indicate the total number of yeast isolates from the total 69 samples.
In line 236, we have corrected the 99 isolates purified from the total number of the samples collected.
Com#2: These are only the purified It is correct? “In total, 99 morphologically distinct isolates were purified from the white-brined 227 cheese production.”
The only purified isolates are 99 in total from all the samples collected.
Minor spelling corrections were applied.
We would be grateful to let us know if any other corrections needed.
Kind regards,
Athina Geronikou
Reviewer 2 Report
In this manuscript, Geronikou and co-authors present a very solid study describing the presence of spoilage yeasts in different sites of dairy production. The study covers more than twenty sites throughout the manufacture of white-brined cheese which includes all relevant processing and processing-related sites (such as air-borne samples). Appropriate methods for genotype classification and identification of the isolates make this study very useful for food biotechnology research. The manuscript is comprehensive and very well written; in this reviewer's eyes, it can be considered for publication in Microorganisms.
Minor issues:
- Lines 56-57: please italicize the genus names.
- The authors should consider defining the "rep-PCR" abbreviation the first time it appears within the text, i.e. line 117 instead of lines 125-126.
Author Response
Dear Editor and Reviewers,
We would like to thank you for your constructive comments on our submitted manuscript ‘Occurrence and identification of yeasts in production of white-brined cheese’ in MDPI Microorganisms Journal.
Regarding the corrections (attached the manuscript with tracking changes):
Com#1: Lines 56-57: please italicize the genus names.
It has been corrected in lines 57-58
Com#2: The authors should consider defining the "rep-PCR" abbreviation the first time it appears within the text, i.e. line 117 instead of lines 125-126.
It has been corrected in lines 120-121.
Minor spelling corrections were applied.
We would be grateful to let us know if any other corrections needed.
Kind regards,
Athina Geronikou